# Insights on the Nutritional Profiling of Cantaloupe (*Cucumis melo* L.) via 1-Naphthalene Acetic Acid

**DOI:** 10.3390/plants12162969

**Published:** 2023-08-17

**Authors:** Sajjad Ali, Atta Ur Rahman, Ehsan Ali, Fadime Karabulut, Saqib Ali, Riaz Ahmad, Mohamed E. Fadl, Mohamed A. E. AbdelRahman, Mohamed A. A. Ahmed, Antonio Scopa

**Affiliations:** 1Department of Botany, Bacha Khan University, Charsadda, Khyber Pakhtunkhwa 24461, Pakistan; sajjadalibkuc@gmail.com (S.A.); att926@gmail.com (A.U.R.); 2Department of Agricultural Chemistry and Biochemistry, The University of Agriculture, Dera Ismail Khan 29220, Pakistan; ehsan.ali@uad.edu.pk; 3Department of Biology, Faculty of Science Firat University, 23119 Elazig, Turkey; karabulutfadime9@gmail.com; 4Department of Entomology, The University of Agriculture, Dera Ismail Khan 29220, Pakistan; saqib.ali@uad.edu.pk; 5Department of Horticulture, The University of Agriculture, Dera Ismail Khan 29220, Pakistan; 6Division of Scientific Training and Continuous Studies, National Authority for Remote Sensing and Space Sciences (NARSS), Cairo 11769, Egypt; adhamnarss@yahoo.com; 7Division of Environmental Studies and Land Use, National Authority for Remote Sensing and Space Sciences (NARSS), Cairo 11769, Egypt; maekaoud@gmail.com; 8Key Laboratory of Agricultural Remote Sensing, Ministry of Agriculture and Rural Affairs, Institute of Agricultural Resources and Regional Planning, Chinese Academy of Agricultural Sciences, Beijing 100081, China; 9Plant Production Department (Horticulture—Medicinal and Aromatic Plants), Faculty of Agriculture (Saba Basha), Alexandria University, Alexandria 21531, Egypt; drmohamedmarey19@alexu.edu.eg; 10School of Agriculture, Yunnan University, Chenggong District, Kunming 650091, China; 11Scuola di Scienze Agrarie, Forestali, Alimentari ed Ambientali (SAFE), Università degli Studi della Basilicata, Via dell’Ateneo Lucano, 10, 85100 Potenza, Italy

**Keywords:** fatty acids, lipids, minerals, nutritious fruit, vitamins, proteins

## Abstract

The nutritional components of cantaloupe, including vitamins, minerals, antioxidants, and dietary fiber, contribute to overall health, improved immunity, hydration, and protection against chronic diseases. This study was conducted to investigate the influence of different concentrations (0 (control), 100, 150, and 200 ppm) of 1-naphthalene acetic acid (1-NAA) on the nutritional components of the cantaloupe (*Cucumis melo* L. Var. Super White Honey). All the studied treatments were applied twice at the 2nd and 4th leaf stages. The applied concentrations of 1-NAA significantly improved the sex expression and fruit yield attributes. Different nutritional components like proximate contents, minerals, vitamins, selected fatty acids, and amino acids were analyzed. The results showed that the maximum moisture content, proteins, carbohydrates, ash, and energy were recorded with 100 ppm. The higher lipids were recorded during the supplementation of 150 ppm. Significantly greater fibers were recorded using 200 ppm. Regarding minerals, 100 ppm was found to be the best as it increased calcium (Ca), magnesium (Mg), potassium (K), sodium (Na), phosphorous (P), manganese (Mn), copper (Cu), iron (Fe), and zinc (Zn). Vitamins were also found to be the maximum with 100 ppm, including vitamin A, vitamin B, vitamin C, vitamin D, vitamin E, and vitamin K. Total selected fatty acids and amino acids were also found significantly greater in the fruits administered 100 ppm.

## 1. Introduction

Cantaloupe, a popular and nutritious fruit, is known for its sweet and refreshing taste. It is rich in essential vitamins, minerals, and antioxidants, making it a valuable addition to a healthy diet. Farmers and researchers are constantly exploring various methods to enhance the nutritional content of crops, and one such approach is the application of plant growth regulators like 1-NAA using foliar sprays [1]. 1-NAA is a synthetic auxin that has been widely used in agricultural practices to promote plant growth and development [2]. When 1-NAA is applied as a foliar spray on cantaloupe plants, it can have a significant impact on their nutritional components [2]. One of the key effects of 1-NAA is its ability to increase fruit size [3]. Several studies have shown that 1-NAA application can lead to larger and heavier cantaloupes, thereby increasing the yield of the crop. With larger fruits, there is a potential for a higher content of essential nutrients [4].

Furthermore, 1-NAA foliar spray has been observed to positively influence the sugar content of cantaloupes [5]. Sugars are the primary source of sweetness in fruits, and by promoting sugar accumulation, 1-NAA can enhance the taste and flavor of the fruit. Moreover, higher sugar content can increase the caloric value of cantaloupes, providing an energy boost to consumers [6]. NAA can also impact the levels of vitamins and antioxidants in cantaloupes. Vitamins, such as vitamin C and β-carotene, play crucial roles in maintaining good health and boosting the immune system. Research suggests that 1-NAA foliar spray can enhance the synthesis and accumulation of these vitamins in cantaloupes, thus increasing their nutritional value [7].

Antioxidants are compounds that protect cells from damage caused by harmful molecules called free radicals. They are associated with numerous health benefits, including reducing the risk of chronic diseases [8]. 1-NAA application has been found to elevate the antioxidant activity in cantaloupes, potentially increasing their capacity to combat oxidative stress and promote overall well-being [9]. 1-NAA foliar spray can have positive effects on the nutritional components of cantaloupes, but the dosage and timing of application are crucial. Excessive or mistimed application of 1-NAA may have negative consequences, such as physiological disorders or compromised fruit quality [10]. Therefore, it is important for farmers to adhere to recommended application rates and follow proper guidelines for optimal results.

Cantaloupe cultivation offers numerous benefits: it is a rich source of essential vitamins and minerals, it offers hydration due to its high water content, and it is a delicious and refreshing addition to a balanced diet [1]. The application of 1-NAA as a foliar spray on cantaloupes can have notable effects on their nutritional components. 1-NAA has the potential to increase fruit size, sugar content, vitamins, and antioxidant activity in cantaloupes, thereby enhancing their overall nutritional value [11].

The study aims to provide evidence-based insights into the use of 1-NAA as a potential strategy to enhance the nutritional quality of cantaloupes. The findings may inform agricultural practices and contribute to the development of sustainable and nutritious crop production techniques.

## 2. Materials and Methods

This study was conducted in the Research Laboratory of the Nuclear Institute for Food and Agriculture (NIFA), Tarnab, Peshawar, Pakistan, in 2018. The site is located at 34°000/N and 71°033/E at an altitude of 400 m above sea level in the Khyber Pakhtunkhwa Province, Pakistan.

### 2.1. Cultivation of Plants

Identified seeds of cantaloupe (cv. Super White Honey) were obtained from Tarnab Farm Peshawar and planted in the last week of February 2018 in plastic pots (11.5 cm long, and 4 cm in diameter) in the greenhouse. Before sowing, the pots were filled with sandy loam containing fertilizer (N, P, and K levels were 0.6, 1.2, and 0.7 g kg^−1^ soil, respectively), and the soil was slightly alkaline (pH 7.3). After seeding, each pot was irrigated with 100 mL of fresh water. During the germination period, the temperature of the greenhouse was about 25 °C to 29 °C. After the first week of seeding, 100% germination was observed. At this stage, 50 mL of water was supplied to each seedling.

### 2.2. Foliar Application of Different 1-NAA Concentrations

1-Naphthalene acetic acid was purchased from PCSIR, Peshawar. Different concentrations of 1-NAA (0, 100, 150, and 200 ppm) were applied to plants. The experiment layout was a randomized complete block design (RCBD) with three replications. The plants were treated with the appropriate amount of 1-NAA solution twice during the vegetative growth stage, i.e., at the 2nd and 4th leaf stages, with the help of a low-pressure hand sprayer. Final harvesting was performed on 28 May 2018.

### 2.3. Observations Recorded

#### 2.3.1. Sex Expression and Yield Attributes of Cantaloupe

Days to first male and female flowerings, number of male and female flowers, and fruit number per vine were also counted. Individual fruit weight and fruit yield per vine were weighed using a digital weighing balance from the thirty tagged plants.

#### 2.3.2. Estimation of Proximate Composition in Cantaloupe

The fruit samples of 5 g were analyzed for moisture content by the oven-dried method. Ash contents were determined by the ignition method as described by Martin [12] and the AOAC [13,14]. Protein was determined by the Kjeldahl method as described by the AOAC [13,14], Sodeke [15], and Khalil and Manan [16]. Fats were determined by the Soxhlet extraction method as described by Hassan et al. [17]. Fibers were determined by the Hennerberg method described by Shumaila and Mahpara [18]. Total carbohydrate content was determined by the “Difference method” as described by Ayoola et al. [19]. Energy values were determined by the standard method as described by Noor et al. [20].

#### 2.3.3. Determination of Minerals in Cantaloupe

K and Na in the sample were determined according to the method of Brenner and Cheikh [21]. P was determined by the described method of the AOAC [13]. Mg, Ca, Cu, Fe, Zn, and Mn were determined by the standard methods of the AOAC [14]. 

#### 2.3.4. Vitamins, Carotenoids, and Fatty Acids Measurements in Cantaloupe

Vitamins in the pulp were determined by titration with standard solutions as described by the AOAC [13,14], Aslam et al. [22], Babarinde et al. [23], and Ringling et al. [24]. Carotenes were determined using the ultraviolet absorption method after extraction with chloroform as described by the AOAC [13,25]. Selected fatty acid contents of the sample were analyzed using the HPLC technique described by Mehta et al. [26].

#### 2.3.5. Determination of Amino Acids in Cantaloupe

Amino acid contents were determined and calculated by acid hydrolysis as described by the AOAC [14]. Methionine, cysteine, and tryptophan are destroyed (because of their heat-sensitive alkyl group) during the process, so these amino acids were not determined.

### 2.4. Statistical Analysis

The collected data of the cantaloupe were analyzed using the analysis of variance (ANOVA) technique under RCBD [27]. Comparison among means was conducted using the LSD test at 0.05 probability via Statistix version 8.1 (Developed by, Cohort software, Berkeley, CA, USA).

## 3. Results

### 3.1. Effect of 1-NAA on Sex Expression and Yield Attributes of Cantaloupe

The foliar spray of 1-NAA at 150 ppm caused the maximum number of days to first male flowering, the number of female flowers per vine, the number of fruits per vine, fruit weight, and fruit yield per vine compared to the other treatments and the control fruit plants. The maximum number of days to first female flowering and the number of male flowers per vine were recorded from the control fruit cantaloupe plants (Table 1). Fruit yield per vine is presented in Figure 1.

### 3.2. Effect of 1-NAA on Proximate Composition of Cantaloupe

Maximum moisture content and energy value were reported from 1-NAA at 100 ppm (Table 2). The foliar spray of 1-NAA at 100 ppm showed significantly higher proteins, lipids, fibers, and ash compared to the other 1-NAA treatments. Carbohydrates were found to be higher using 1-NAA at 150 ppm than at the other applied concentrations (Table 2).

### 3.3. Effect of 1-NAA on Nutrients and Minerals (Macro- and Micronutrients) of Cantaloupe

The exogenous spray of 1-NAA at 100 ppm led to a higher increase in the endogenous level of macronutrients such as P, Ca, K, Mg, and Na compared to the other treatments and control fruits (Table 3). The supplementation of 1-NAA at 100 ppm enhanced the endogenous level of micronutrients such as Mn, Cu, Fe, and Zn, while decreased levels of all the studied micronutrients were recorded in the control fruit (Table 4).

### 3.4. Effect of 1-NAA on Different Vitamins in Cantaloupe

The maximum vitamin B-12 content was recorded during the spray of 1-NAA at 100 ppm. The significantly higher thiamine, niacin, vitamin C, vitamin E, vitamin A, and vitamin K levels were recorded during the application of 1-NAA at 100 ppm. However, riboflavin remained non-significant among all the studied treatments of 1-NAA (Table 5). Moreover, vitamin D was also found to be higher in 1-NAA at 100 ppm (Figure 2).

### 3.5. Effect of 1-NAA on Carotenes in Cantaloupe

Carotenes (α carotene and β carotene) were significantly improved in the cantaloupe with the supplementation of 1-NAA. The maximum increase in the studied carotenes was recorded during 1-NAA at 100 ppm, while the minimum was measured in the control fruits (0 ppm of 1-NAA), which was equal to the value obtain from 1-NAA at 200 ppm (Figure 3).

### 3.6. Effect of 1-NAA on Different Amino Acids in Cantaloupe

All the studied amino acids were significantly improved in the cantaloupe with the supplementation of 1-NAA. Different amino acids, i.e., arginine, alanine, aspartic acid, glutamic acid, glycine, proline, histidine, serine, valine, tyrosine, and phenylalanine, showed a higher increase under the supplementation of 1-NAA at 100 ppm compared to the other studied treatments of 1-NAA. All the studied amino acids were at low levels in those plants in which 1-NAA application was missing (Table 6). The maximum amino acid concentrations (lysine, leucine, and isoleucine) were recorded during the application of 1-NAA at 100 ppm, while the minimum carotenes were measured in the control fruits (1-NAA at 0 ppm) (Figure 4).

### 3.7. Effect of 1-NAA on Fatty Acids in Cantaloupe

All the studied fatty acids were enhanced in the Cantaloupe administered the pre-harvest exogenous application of 1-NAA. The increased concentration of 1-NAA can improve the concentration of fatty acids in the treated fruits. The maximum fatty acids were recorded during the application of 1-NAA at 100 ppm (Table 7). Moreover, the calibration data are illustrated for the standard solutions in the fatty acid determination procedure in the cantaloupe (Table 8).

## 4. Discussion

1-NAA is a plant growth regulator commonly used in agriculture. When applied to cantaloupe crops, 1-NAA can influence sex expression and yield. Promoting female flower development enhances the fruit set and increases yield potential. 1-NAA stimulates ethylene production, leading to the development of more female flowers, which are essential for fruit formation [28]. Additionally, 1-NAA can improve fruit quality by increasing sugar content and enhancing fruit size. In the present study, 150 ppm was found to be effective for a higher yield. However, it is crucial to carefully regulate the dosage and timing of 1-NAA application as excessive use can result in negative effects such as malformed fruit or a reduced yield [29]. The proper management of 1-NAA can optimize sex expression and yield in cantaloupe crops. Our findings are also confirmed by the work of Hidayatullah et al. [28] on bottle gourd, Vadigeri et al. [30] on cucumber, and Baset Mia et al. [29] on bitter gourd. 1-NAA improved cell division and elongation, which further results in higher fruit weight and size. Moreover, its application is very effective for increasing shelf life under storage as it reduces ethylene production, which restricts senescence in fruits [23].

Proximate composition refers to the measurement of various components such as moisture, energy, protein, fat, fiber, ash, and carbohydrates in a food sample. In this study, the results showed that the application of 1-NAA significantly influenced the proximate composition of cantaloupe. Specifically, it was found that the treatment with 1-NAA resulted in increased levels of protein, fiber, lipids, ash, and carbohydrates, while the levels of moisture and energy remained relatively unchanged. These findings suggest that the application of 1-NAA can enhance the nutritional composition of cantaloupe, particularly in terms of protein, lipids, and carbohydrate content [31]. The current research explored the underlying mechanisms and potential applications of 1-NAA in improving the proximate composition of cantaloupe. The results are close to the findings of Reenata et al. [31], Arora and Prata [32], Stankovic et al. [33], and Reenata et al. [31], who also studied the effect of NAA on such parameters.

1-NAA is a synthetic plant hormone widely used in agriculture to enhance fruit growth and development [34]. It had a significant impact on macronutrients and micronutrients when applied to cantaloupe. Macronutrients like N, P, Ca, Mg, and K play crucial roles in plant growth, and 1-NAA can enhance their uptake and utilization by the fruit. This results in improved fruit size, color, and quality [1]. Furthermore, 1-NAA also influences the uptake and distribution of micronutrients such as Mn, Fe, Zn, and Mn in cantaloupe. These micronutrients are essential for various physiological processes and contribute to the nutritional value of the fruit. By promoting their absorption and translocation, 1-NAA helps to maintain optimum levels of micronutrients in the fruit, enhancing its nutritional profile [35]. Overall, the application of 1-NAA on cantaloupe positively affects the macronutrient and micronutrient composition, leading to improved fruit quality and enhanced nutritional value. The results of the present investigation are close to the observations of Jacob et al. [35] and Ajay et al. [36]. A significant increase in the mineral contents was probably due to the increase in the permeability of the cantaloupe root hairs, which allowed the fruit to absorb the maximum amount of dissolved minerals from the soil by stretching the cell walls of the root cells [36].

The impact of 1-NAA on the vitamin content of cantaloupe largely depends on the concentration and timing of application [37]. According to the present results, 100 ppm of 1-NAA significantly improves the vitamin contents in cantaloupe. The current results align with the results of earlier research that showed that 1-NAA treatments can increase the levels of certain vitamins, such as vitamin C, vitamin A, and vitamin E, in cantaloupes [38]. However, the effects on other vitamins, like the vitamin B complex and vitamin K, may vary or even be negligible. It is important to carefully consider dosage and application methods to optimize the desired vitamin enhancement in cantaloupe. So, 100 ppm of 1-NAA is effective for enhanced vitamins in cantaloupe. The results are similar to those reported in the investigations of Morais et al. [34] and Mata et al. [37].

In cantaloupe, the application of 1-NAA has shown significant effects on different carotenes, which are responsible for the vibrant coloration and nutritional value of the fruit. 1-NAA stimulates the synthesis of carotenes such as beta-carotene, lycopene, and lutein, resulting in an increased accumulation of these pigments in cantaloupe. So, similar findings were revealed from the present results. This leads to enhanced color development, making the fruits more visually appealing. Additionally, the higher carotene content contributes to improved nutritional quality as these compounds possess antioxidant properties and are beneficial for human health. The increase in the carotenoid contents following the application of 1-NAA at 100 ppm is probably related to more uptakes of inorganic mineral substances from the soil [34].

The application has a significant impact on the amino acid composition of cantaloupe. 1-NAA applied to cantaloupe significantly influenced the biosynthesis and accumulation of amino acids [30]. Studies have shown that 1-NAA treatment leads to an increase in essential amino acids such as lysine, threonine, and valine, which are crucial for human nutrition. These amino acids play a vital role in protein synthesis and various metabolic processes. 1-NAA treatment has also been found to enhance the content of non-essential amino acids like glutamic acid and aspartic acid [34]. The effect of 1-NAA on amino acids in cantaloupe highlights its potential to improve the nutritional value and flavor profile of the fruit. These findings contribute to our understanding of the physiological responses of plants to growth regulators and aid in optimizing fruit production techniques. These observations are close to those of the investigation of Annalisa et al. [33]. The application of 100 ppm probably enhanced the potential of roots to absorb the maximum amount of minerals and leaves to gain higher CO_2_ from the air. It seems that the 100 ppm treatment also increased the already present amino acids’ metabolism.

Research has shown that the application of 1-NAA can influence the fatty acid profile of cantaloupe [28]. It has been revealed that 1-NAA treatment can increase the levels of certain unsaturated fatty acids, such as linoleic acid and oleic acid, which are considered beneficial for human health. This suggests that 1-NAA application has the potential to enhance the nutritional value of cantaloupe by promoting the accumulation of desirable fatty acids [34]. However, further studies are required to determine the optimal concentration and timing of 1-NAA application for maximizing these effects while considering potential negative impacts. This result was close to the findings of Rathod et al. [39] and Essien et al. [40]. According to the WHO, the recommended daily intakes for adults aged 20–70 are 11,000–17,000 mg for linoleic acid and 1100–1600 mg for linolenic acid. This study has shown that cantaloupe can fulfill the need for essential fatty acids very well.

## 5. Conclusions

In conclusion, the application of 1-NAA has proven to be a promising approach for enhancing the nutritional components of cantaloupe. Using 1-NAA, key nutritional elements such as vitamins, minerals, and antioxidants can be significantly increased, leading to improved nutritional value and potential health benefits for consumers. Overall, enhancing the nutritional components of cantaloupes via 1-NAA holds great potential for improving the nutritional profile and value of this popular fruit, thereby contributing to healthier food options and promoting well-being in the future.

## Figures and Tables

**Figure 1 plants-12-02969-f001:**
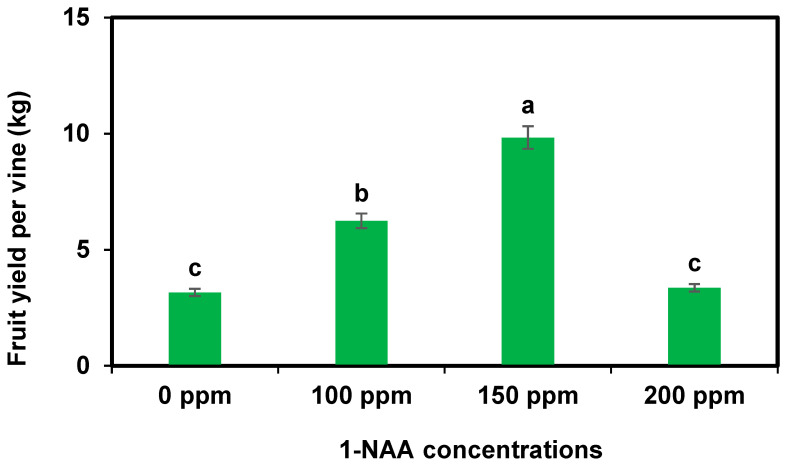
Effect of 1-NAA on fruit yield per vine of cantaloupe crop. Mean values with different letters (s) in a column are statistically significant at *p* ≤ 0.05 (LSD test).

**Figure 2 plants-12-02969-f002:**
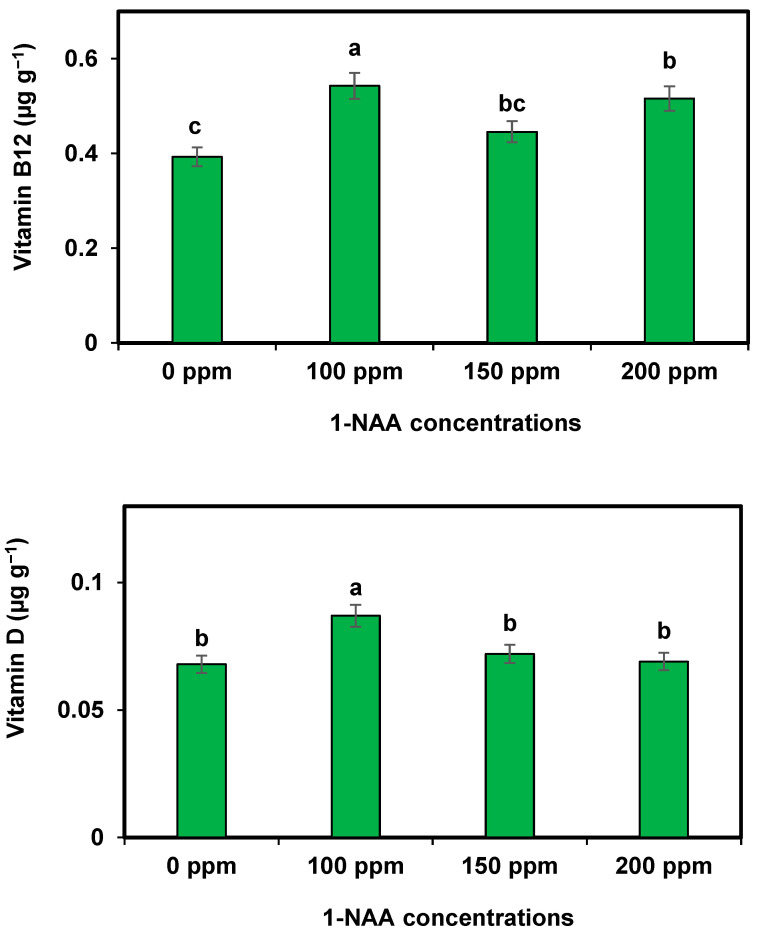
Effect of 1-NAA on vitamin B12 and vitamin D concentrations in cantaloupe. Mean values with different letters (s) in a column are statistically significant at *p* ≤ 0.05 (LSD test).

**Figure 3 plants-12-02969-f003:**
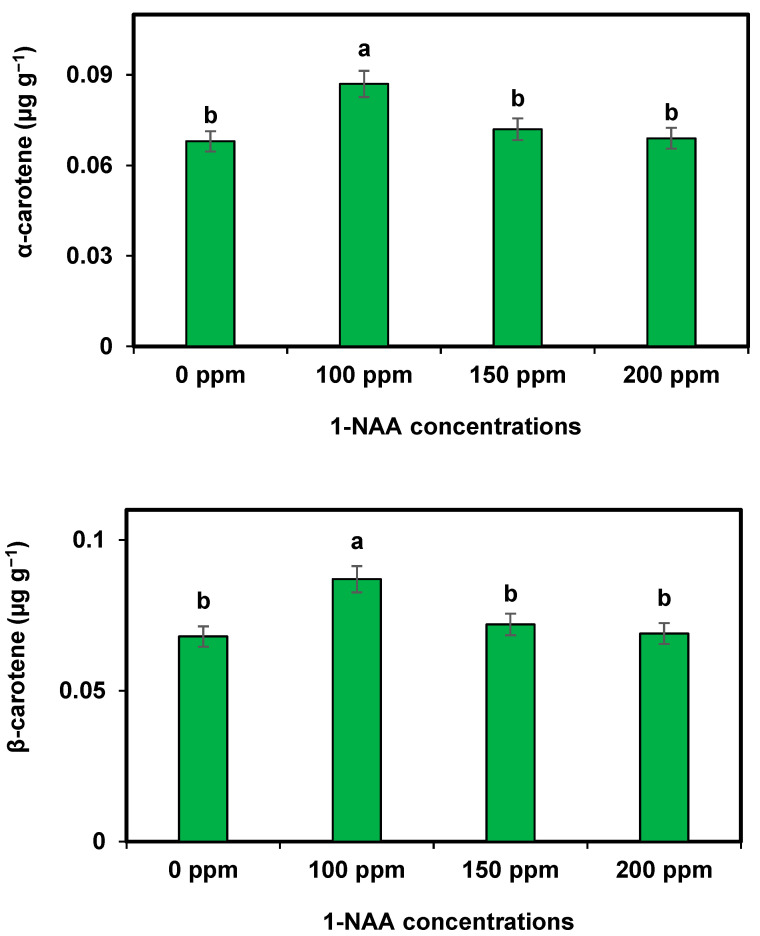
Effect of 1-NAA on carotenes of cantaloupe. Mean values with different letters (s) in a column are statistically significant at *p* ≤ 0.05 (LSD test).

**Figure 4 plants-12-02969-f004:**
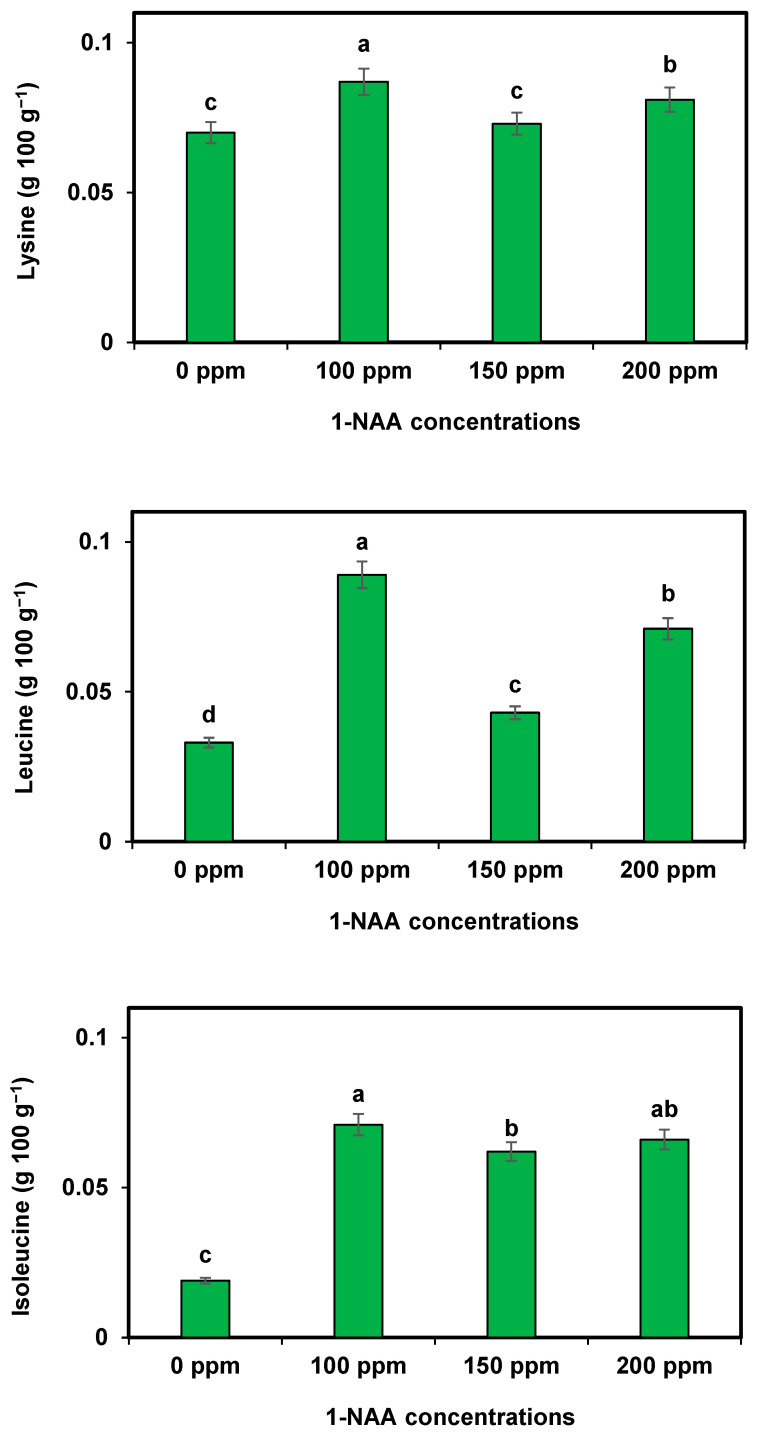
Effect of 1-NAA on amino acids (lysine, leucine, and isoleucine) of cantaloupe. Mean values with different letters (s) in a column are statistically significant at *p* ≤ 0.05 (LSD test).

**Table 1 plants-12-02969-t001:** Effect of 1-NAA on sex expression and yield attributes of cantaloupe.

Treatment Applications	Days to First Male Flowering	Days to First Female Flowering	Number of Male Flowers per Vine	Number of Female Flowers per Vine	Number of Fruits/Vines	Fruit Weight (kg)
1-NAA at 0 ppm (control)	17.56 ± 0.73 c	48.09 ± 0.61 a	118.31 ± 0.76 a	9.21 ± 0.53 b	3.73 ± 0.44 c	0.83 ± 0.56 b
1-NAA at 100 ppm	27.13 ± 0.76 b	28.04 ± 0.58 b	76.06 ± 0.74 b	10.27 ± 0.52 b	5.92 ± 0.45 b	1.04 ± 0.57 ab
1-NAA at 150 ppm	34.76 ± 0.81 a	17.51 ± 0.57 c	55.17 ± 0.70 c	13.93 ± 0.55 a	7.79 ± 0.47 a	1.21 ± 0.58 a
1-NAA at 200 ppm	19.71 ±0.71 c	11.03 ± 0.55 d	66.81 ± 0.69 d	7.11 ± 0.50 c	3.43 ± 0.43 c	0.92 ± 0.51 b

Mean values sharing different lettering showed significant effects on the studied traits at 0.05 probability.

**Table 2 plants-12-02969-t002:** Effect of 1-NAA on comparative analysis of proximate composition (g/100 g fresh weight) of cantaloupe.

Treatment Applications	Moisture	Energy	Protein	Lipids	Fibers	Ash	Carbohydrates
1-NAA at 0 ppm (control)	93.3 ± 0.71 a	32.1 ± 0.51 a	11.6 ± 0.62 b	1.8 ± 0.77 c	12.2 ± 0.47 b	6.1 ± 0.72 b	10.6 ± 0.47 b
1-NAA at 100 ppm	97.2 ± 0.70 a	36.3 ± 0.53 a	15.3 ± 0.64 a	2.7 ± 0.81 a	16.1 ± 0.53 a	14.4 ± 0.76 a	8.3 ± 0.45 c
1-NAA at 150 ppm	95.1 ± 0.67 a	33.1 ± 0.52 a	12.2 ± 0.63 b	2.9 ± 0.80 a	13.3 ± 0.48 b	7.3 ± 0.73 b	12.2 ± 0.49 a
1-NAA at 200 ppm	94.2 ± 0.66 a	32.7 ± 0.51 a	12.7 ± 0.63 b	2.2 ± 0.79 b	16.3 ± 0.54 a	6.5 ± 0.70 b	10.4 ± 0.47 b

Mean values sharing different lettering showed significant effects on the studied traits at 0.05 probability.

**Table 3 plants-12-02969-t003:** Effect of 1-NAA on macronutrients (mg/100 g fresh weight) of cantaloupe.

Treatment Applications	P	Ca	K	Mg	Na
1-NAA at 0 ppm (control)	93.4 ± 0.54 b	165.3 ± 0.70 b	297.6 ± 0.81 d	130.9 ± 0.70 d	118.1 ± 0.59 d
1-NAA at 100 ppm	106.1 ± 0.58 a	180.3 ± 0.73 a	3821.2 ± 0.87 a	145.3 ± 0.78 a	154.3 ± 0.63 a
1-NAA at 150 ppm	96.3 ± 0.56 b	170.2 ± 0.71 b	3644.7 ± 0.86 b	135.2 ± 0.73 c	122.6 ± 0.61 c
1-NAA at 200 ppm	97.2 ± 0.57 b	168.2 ± 0.68 ab	3401.3 ± 0.84 c	140.1 ± 0.75 b	137.2 ± 0.64 b

Mean values sharing different lettering showed significant effects on the studied traits at 0.05 probability.

**Table 4 plants-12-02969-t004:** Effect of 1-NAA on micronutrients (mg/100 g fresh weight) of cantaloupe.

Treatment Applications	Mn	Cu	Fe	Zn
1-NAA at 0 ppm (control)	1.5 ± 0.49 d	1.0 ± 0.32 d	4.1 ± 0.53 c	3.1 ± 0.55 d
1-NAA at 100 ppm	2.7 ± 0.52 a	1.7 ± 0.36 a	7.3 ± 0.57 a	6.4 ± 0.59 a
1-NAA at 150 ppm	2.2 ± 0.51 b	1.3 ± 0.33 c	5.5 ± 0.54 b	6.0 ± 0.58 ab
1-NAA at 200 ppm	1.9 ± 0.50 c	1.5 ± 0.35 b	4.0 ± 0.53 c	3.7 ± 0.56 c

Mean values sharing different lettering showed significant effects on the studied traits at 0.05 probability.

**Table 5 plants-12-02969-t005:** Effect of 1-NAA on different vitamins (mg/g fresh weight) of cantaloupe.

Treatment Applications	Vitamin B-12	Thiamine	Riboflavin	Niacin	Vitamin C	Vitamin E	Vitamin A	Vitamin K
1-NAA at 0 ppm (control)	0.67 ± 0.70 c	0.052 ± 0.30 b	0.023 ± 0.34 a	0.81 ± 0.42 c	53.64 ± 0.50 c	0.156 ± 0.32 d	56.33 ± 0.64 c	3.06 ± 0.56 c
1-NAA at 100 ppm	1.14 ± 0.78 a	0.082 ± 0.34 a	0.047 ± 0.38 a	1.29 ± 0.47 a	64.72 ± 0.55 a	0.213 ± 0.38 a	62.21 ± 0.70 a	4.56 ± 0.49 a
1-NAA at 150 ppm	0.71 ± 0.73 c	0.064 ± 0.33 b	0.033 ± 0.36 a	1.11 ± 0.46 a	55.34 ± 0.54 b	0.209 ± 0.36 b	58.00 ±0.68 b	3.35 ± 0.47 b
1-NAA at 200 ppm	1.03 ± 0.76 b	0.057 ± 0.32 b	0.027 ± 0.35 a	0.92 ± 0.44 b	56.93 ± 0.53 b	0.179 ± 0.35 c	55.36 ± 0.65 c	3.06 ± 0.56 c

Mean values sharing different lettering showed significant effects on the studied traits at 0.05 probability.

**Table 6 plants-12-02969-t006:** Effect of 1-NAA on different amino acids (g/100 g fresh weight) of cantaloupe.

Treatment Applications	Arginine	Alanine	Aspartic acid	Glutamic Acid	Glycine	Proline	Histidine	Serine	Valine	Tyrosine	Phenylalanine
1-NAA at 0 ppm (control)	0.031 ± 0.80 d	0.092 ± 0.35 d	0.134 ± 0.48 d	0.213 ± 0.60 c	0.029 ± 0.70 c	0.022 ± 0.43 d	0.011 ± 0.47 c	0.038 ± 0.72 c	0.029 ± 0.50 c	0.017 ± 0.39 c	0.027 ± 0.30 d
1-NAA at 100 ppm	0.114 ± 0.85 a	0.189 ± 0.41 a	0.177 ± 0.53 a	0.267 ± 0.65 a	0.044 ± 0.74 a	0.064 ± 0.50 a	0.034 ± 0.53 a	0.112 ± 0.81 a	0.057 ± 0.56 a	0.036 ± 0.49 a	0.097 ± 0.38 a
1-NAA at 150 ppm	0.088 ± 0.82 c	0.134 ± 0.39 b	0.156 ± 0.50 c	0.216 ± 0.61 c	0.039 ± 0.72 b	0.061 ± 0.47 b	0.025 ± 0.49 b	0.079 ± 0.74 d	0.045 ± 0.53 b	0.027 ± 0.44 b	0.036 ± 0.33 c
1-NAA at 200 ppm	0.096 ± 0.83 b	0.122 ± 0.37 c	0.161 ± 0.51 b	0.221 ± 0.63 b	0.041 ± 0.73 a	0.053 ± 0.45 c	0.027 ± 0.51 b	0.103 ± 0.78 b	0.051 ± 0.55 a	0.031 ± 0.47 a	0.071 ± 0.36 b

Mean values with different letters (s) in a column are statistically significant at *p* ≤ 0.05 (LSD test).

**Table 7 plants-12-02969-t007:** Effect of 1-NAA on fatty acids (mg/ 100 g fresh pulp) of cantaloupe.

Fatty Acids	1-NAA at 0 ppm (Control)	1-NAA at 100 ppm	1-NAA at 150 ppm	1-NAA at 200 ppm
14:0	2.6	3.2	2.9	2.7
16:0	114.8	122.2	118.0	115.3
17:0	ND	ND	ND	ND
18:0	14.6	18.2	14.9	16.1
20:0	3.4	4.7	4.2	4.2
20:in-9	0.6	0.9	0.7	0.8
22:0	0.3	0.7	0.4	0.5
22:6n3	0.6	0.9	0.7	0.7
24:0	3.7	4.1	3.9	3.8
16:in-9	8.7	9.5	8.8	9.0
18:in-9	22	26.2	24.9	24
18:in-7	13.2	17.4	14.1	14.7

**Table 8 plants-12-02969-t008:** Calibration data for the standard solutions in fatty acids determination procedure.

Fatty Acids (Phenacyle Bromide)	Linearity Interval (mg L^−1^)	Determination Coefficient (r^2^)
Linoleic acid	98–384	0.9917
Linolenic acid	17–67	0.9934
Mystiric acid	0.19–2.13	0.9976
Oleic acid	77–296	0.9658
Palmatic acid	0.244–6.33	0.9367
Stearic acid	0.273–5.82	0.9497

## Data Availability

Not applicable.

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
