# Peer review of "Insights on the Nutritional Profiling of Cantaloupe (Cucumis melo L.) via 1-Naphthalene Acetic Acid"

_plants, 2023, doi:10.3390/plants12162969_

Round 1
Reviewer 1 Report
1. Although this manuscript is not so original, it is broad, sufficiently supported and interesting. However, it presents some aspects that must be reviewed and addressed in order for it to be accepted for publication. Below are the main observations that I consider should be addressed.
2. The title alludes to enlightening climate resilience. However, in no other section of the manuscript, this is not even mentioned. It is suggested to change this title to another more appropriate one.
3. In section 2.1 Cultivation of plants:
About halfway through the paragraph it says: “…, the soil was slightly saline (pH 7.3).”. I believe that if the soil pH was 7.3, this indicates that this soil was slightly alkaline, but not slightly saline. It is suggested to correct the data.
4. In the section 2.3.4. Determination of vitamins, carotenoids, and fatty acids of cantaloupe fruits
Near the end of the paragraph it says: “Carotenes were determined through the ultraviolet absorption method after extraction with chloroform as described by Duncan [25]”.
I consider that this reference does not correspond to any method of absorption or extraction, but to a classic treatise on statistical methodology.
5. In section 3.1. Effect of 1-NAA on sex expression and yield attributes of cantaloup fruits
It says: “Foliar spray of 1-NAA (150 ppm) revealed the maximum number of days to first male flowering”. I consider that the verb used (to reveal) is incorrect. I think it should say: “Foliar spray of 1-NAA (150 ppm) caused the maximum number of days to first male flowering”.
6. Throughout the results section, the data shown in Tables 1, 2, 3, 4, 5, and 6 appear unnecessarily repeated in the text. It is suggested to avoid this repetition of data.
7. Table 2 does not indicate in which units the Moisture and Energy data are expressed.
8. In section 3.4. Effect of 1-NAA on different vitamins in cantaloup fruits:
it says: “The significantly higher thiamine (0.082 mg/ g), niacin (1.29 mg/ g), vitamin C (64.72 mg/ g), vitamin E (0.213 mg/ g), vitamin A (62.21 mg/ g), and vitamin K (4.56 mg/ g) were recorded during application of 1-NAA (100 ppm)”. I think it should say: “The significantly higher thiamine, niacin, vitamin C, vitamin E, vitamin A, and vitamin K were recorded after application of 1-NAA (100 ppm)”. The same observation for the section 3.7.
Author Response
Dear Reviewer,
I am writing to submit the revised version of my manuscript titled “Enlightening the nutritional profiling of cantaloupe (Cucumis melo L) via 1-naphthalene acetic acid” (Plants, MDPI). I would like to express my gratitude for the valuable feedback and constructive suggestions provided by the reviewers. Their insights have been instrumental in enhancing the quality and clarity of my work.

Reviewer 2 Report
The manuscript is well prepared and presented. Below you can find more detailed comments.
- Please provide the appropriate references for any claim you are making in the introduction and discussion sections e.g., “Farmers and researchers are constantly exploring various methods […] through foliar sprays”, “When 1-NAA is applied as a foliar spray […] on their nutritional components”, lines 15-17 etc.
- The last sentence of the introduction reads as a conclusion. Please revise.
- What was the sample size used for the analyses? How many biological and technical replicates were performed?
- Please provide the SD values for the data presented on Tables 1 - 7. Also, is it mg/ 100 g fresh weight or dry weight?
- When citing a published study, you should also mention the year of the publication.
- Some of the cited references are quite old. Please try to use more recent literature.
The language needs some refinement.
Author Response

(The authors gave the same response as above.)

Reviewer 3 Report
I have reviewd the manuscript with the title: Enlightening climate resilience and nutritional profiling of cantaloupe (Cucumis melo L) using 1-naphthalene acetic acid
General comments:
-The line numbers are missing so it is difficult to review.
- why did the authors measerue fatty acids which give only a little part to the overal fruit quality ?
- english need major revisons...
Abstract: After a successful life cycle, fruits - this sentence is not necesarry in the abstract.
Keywords: all should be written the same, now one word is with capital letter and other with no capital letters...
Materials and methods.
The titles of each paragraph should be revised since there are errors.
Please provide from where the 1-NAA was purchased.
Check english in this section since it is poor.
The statistical analysis section is poorly written. It should be described in more details.
Results
The yield data should be in a graph, not a table, so they are more clear.
Disscusion
It seems that 100 also inc... This should be written It seems that the 100 ppm treatment...
The english should be revised, since now it is in a poor condition.
Author Response

(The authors gave the same response as above.)

Reviewer 4 Report
The studies carried out by the authors on the effect of spraying with growth regulator 1-naphthalene acetic acid on cantaloupe (Cucumis melo L.) are relevant and practical. Modern research methods are used. The article may be published after some revision.
1 1. How does the article show the climate change resilience study of cantaloupe (Cucumis melo L.)? The title of the article needs to be changed.
2. Please indicate the purpose of the study in the Introduction.
3. Section "3.4. Effect of 1-NAA on different vitamins in cantaloup fruits". According to Figure 1, the lower vitamin B12 and vitamin D were recorded not only in control plants. The statistics show slightly different data.
4. Section «3.5. Effect of 1-NAA on carotenes in cantaloupe fruits». Statistical data is not quite correctly interpreted. The minimum carotenes were measured not only in the control fruits 1-NAA (0 ppm). The content of carotenes (α carotene and β carotene) differed statistically significantly only when treated with 1-NAA (100 ppm). The level of carotenoids when treated with other concentrations of 1-NAA remains at the level of the control.
5. How does 1-NAA spraying affect fruit weight and storage?
Author Response

(The authors gave the same response as above.)

Round 2
Reviewer 2 Report
The manuscript has improved significantly after the revision. However, there are still a couple of changes that are required prior to publication. Please see below.
- The title doesn’t make much sense; do you mean enhancing? Please revise.
- Please add the year of the publication in all the cited literature in the main text of the manuscript; lines 26, 27, 42, 56, 68, 101.
Language still needs some minor editing. However, I believe that the Journal provides editing services prior to publication.
Author Response
The authors are very grateful for the suggestions received from the Referees and the Editor. We thank you for the work you have done which has led to our manuscript being publishable in the prestigious journal Plants-MDPI.
Relatively to
- The title doesn't make much sense; do you mean enhancing? Please revise.
We've tried changing the title and hope it's more appropriate and makes more sense.
- Please add the year of the publication in all the cited literature in the main text of the manuscript; lines 26, 27, 42, 56, 68, 101.
Unfortunately, we were unable to find, in the lines indicated, where to insert the year of publication in the text of the manuscript.
With regard to the English language we believe, as you advised, that small changes and errors will be corrected by the editorial service.
Thanks again
Best regards
Prof. Antonio Scopa
Reviewer 3 Report
The autohrs have addresed all my comments.
The english language is better.
Author Response
Dear reviewer
Thanks to you, the other reviewers, and the editor for helping us make our manuscript publishable.
Best regards
Prof. A. Scopa
Reviewer 4 Report
Dear authors!
Congratulations. The article is well edited.
Author Response

(The authors gave the same response as above.)
